# Identification of Genetic Risk Factors for Keratinocyte Cancer in Immunosuppressed Solid Organ Transplant Recipients: A Case-Control Study

**DOI:** 10.3390/cancers15133354

**Published:** 2023-06-26

**Authors:** Raute Sunder-Plassmann, Alexandra Geusau, Georg Endler, Wolfgang Weninger, Matthias Wielscher

**Affiliations:** 1Department of Laboratory Medicine, Medical University of Vienna, 1090 Vienna, Austria; raute.sunder-plassmann@meduniwien.ac.at (R.S.-P.);; 2Department of Dermatology, Medical University of Vienna, 1090 Vienna, Austria; alexandra.geusau@meduniwien.ac.at (A.G.);

**Keywords:** keratinocyte cancer, solid organ transplant recipients, immunosuppression, whole exome sequencing

## Abstract

**Simple Summary:**

Solid organ transplant recipients are at increased risk for keratinocyte cancer of the skin. In our study, we carefully matched our cases and controls for major risk factors of keratinocyte cancer such as age, sex, transplanted organ, post-transplant period, immunosuppressive therapy, and UV exposure. In addition, they also had comparable cumulative UV exposure. Using this approach, we identified several genetic loci involved in pigmentation/UV protection, tumor suppression, immunomodulation, intracellular traffic, and response to UV associated with the occurrence of multiple keratinocyte cancers.

**Abstract:**

Because of long-term immunosuppression, solid organ transplant recipients are at increased risk for keratinocyte cancer. We matched solid organ transplant patients (*n* = 150), cases with keratinocyte cancers and tumor-free controls, considering the most important risk factors for keratinocyte cancer in solid organ transplant recipients. Using whole exome data of germline DNA from this patient cohort, we identified several genetic loci associated with the occurrence of multiple keratinocyte cancers. We found one genome-wide significant association of a common single nucleotide polymorphism located in *EXOC3* (rs72698504). In addition, we found several variants with a *p*-value of less than 10^−5^ associated with the number of keratinocyte cancers. These variants were located in the genes *CYB561*, *WASHC1*, *PITRM1-AS1*, *MUC8*, *ABI3BP,* and *THBS2-AS1*. Using whole exome sequencing data, we performed groupwise tests for rare missense variants in our dataset and found robust associations (*p* < 10^−6^, Burden Zeggini test) between *MC1R*, *EPHA8*, *EPO*, *MYCT1*, *ADGRG3*, and *MGME1* and keratinocyte cancer. Thus, overall, we detected genes involved in pigmentation/UV protection, tumor suppression, immunomodulation, intracellular traffic, and response to UV as genetic risk factors for multiple keratinocyte cancers in solid organ transplant recipients. We also grouped selected genes to pathways and found a selection of genes involved in the “cellular response to UV” to be significantly associated with multiple keratinocyte cancers.

## 1. Introduction

Because of long-term immunosuppression, solid organ transplant recipients (SOTR) are at increased risk for keratinocyte cancer (KC), making their tumor development an ideal model for skin cancer [1,2]. The pathogenesis of keratinocyte cancer (KC) involves several mechanisms including exposure to ultraviolet radiation (UVR), which is the main cause of DNA damage in keratinocytes [3,4]. In a cohort of 388 SOTR (matched for age, sex, gender, transplanted organ, post-transplant (TX) period, and immunosuppressive (IS) therapy), we previously evaluated the risk for KC. Skin aging, UV-related skin damage, or cumulative UV exposure could not explain KC development. Only holiday-related high-intensity UV exposure, specific *MC1R* variants, and the presence of actinic keratosis (AK) increased the risk for KC [5]. Thus, we think, that additional, probably genetic, factors may predispose to the development of KC in this thoroughly characterized population of SOTR.

From a genetic point of view, KC is a polygenic disease. Many oncogenes, tumor suppressor genes, DNA repair genes, and genes modulating skin pigmentation are likely to be involved. Rare alleles with high penetrance and common alleles with low penetrance probably contribute to genetic susceptibility to cutaneous squamous cell carcinoma (cSCC) and basal cell carcinoma (BCC). These are the most common malignancies after solid organ TX, accounting for more than 90% of all skin cancers in these patients [6,7,8]. KC risk assessment including the genetic stratification of SOTR is crucial for identifying patients who would benefit from a more vigorous dermatologic surveillance.

Our aim was to understand genetic factors that increase KC risk and thus explain KC development, which was not clarified by our previous studies in this patient cohort [5]. To achieve this, we evaluated the contribution of common variants on the one hand and rare variants on the other hand. We evaluated rare variants grouped by gene and several genes grouped into panels to gain more insight into possible pathways involved in KC development.

## 2. Patients and Methods

### 2.1. Patients

In our prospective case control study, we included 150 SOTR (cases/controls, *n* = 75 each) recruited between 2017 and 2019. The cases had at least five histologically proven KCs (cSCC, BCC, Bowen’s disease (BD), and Bowen’s carcinoma (BowCa)) after TX. Cases with a tumor burden of ≥5 and up to 130 KCs per individual were matched with tumor-free controls for the known KC risk factors of gender, age, type of transplanted organ, post-TX period, and immunosuppressive (IS) therapy. In addition, they also had comparable values for cumulative UV exposure. In order to determine the likelihood of discovering SNPs associated with KC in our cohort, we conducted a power analysis. This analysis aimed to estimate the minimum sample size required to achieve a 90% probability (i.e., power) of confidently detecting SNPs associated with KC. Given the careful matching of cases and controls in our relatively small cohort, we had to assume relatively high effect sizes. For the alternative hypothesis, we assumed an effect size of 0.75, indicating the magnitude of the association between an SNP and the development of KC. The alpha level, which represents the significance level of the associated SNP in the analysis, was set at the genome-wide significance level of 5 × 10^−8^. To obtain estimates for the required sample sizes to achieve 90% power, we performed simulations by varying the minor allele frequencies from 0.1 to 0.5. The results of our simulations, as shown in Appendix A, indicated that a study size of 150 samples would achieve 90% power for minor allele frequencies as low as 0.2.

### 2.2. Whole Exome Analysis

Genomic DNA was isolated from the patients’ peripheral blood using the QIAsymphony (Qiagen, Hilden, Germany) according to the manufacturer’s recommendations. Next generation sequencing/WES was performed by Sophia Genetics (Saint Sulpice, Switzerland). Briefly, target enrichment was performed with a custom designed Agilent SureSelect panel including 203,058 target regions distributed over 19,682 genes spanning a total of 40,907,213 bp of the human genome. Paired End NGS libraries were sequenced on an Illumina NovaSeq sequencer (Illumina, San Diego, CA, USA). The average read length was 150 bp. The target sequencing depth of enriched genomic regions was 100×.

### 2.3. Alignment and Variant Calling

Here, 150 bp paired end reads were aligned against GRCh38 using Smith-Waterman Alignment scoring, as implemented by Illumina’s Dragen pipeline (Illumina, San Diego, CA, USA). Duplicate reads were marked; reads were also soft-trimmed before alignment. An overview of the raw read processing procedure is given in Appendix A. Variants were called using Dragen Variant Caller Algorithm. Additionally, we estimated the ploidy, cross contamination, presence in dbSNP database, Ti/Tv ratio, and heterozygous to homozygous ratio (Appendix A). To generate a unified call set consisting of 150 samples, each individual call set was restricted to variants with Filter PASS as assigned by the Dragen variant caller. The Dragen PASS flag is based on a Machine Learning recalibration procedure based on the available read and context information such as surrounding haplotypes and general variant calling QC metrics such as strand bias, sequencing depth, etc. Variants were normalized and left aligned. We used the bcftools merge function to combine all VCF files to one call set. Missing genotypes were set to 0/0. We used plink software [9] to calculate the principal components, where the minimum minor allele frequency was set as 5% with a Hardy−Weinberg Equilibrium *p*-value of no less than 10^−4^. Variant annotation was performed with the Ensembl Variant effect predictor (VEP) applying the VEP-everything flag [10].

### 2.4. Association Testing

We performed group wise burden and kernel tests, as well as single score tests using rvtests software [11]. In our regression models, the number of KC was used as a continuous outcome. We adjusted the regression models with the following covariates: age, sex, and the first six principal components as inferred from the dataset using plink software. For score tests (one regression per variant), the data set was restricted to variants with a minimum minor allele frequency of 5% in this dataset (*n* = 81,919). Summary statistics of this regression are available here https://doi.org/10.6084/m9.figshare.22494538.v1 accessed on 20 June 2023 (link to figshare). To evaluate the association between all rare variants per gene and KC, we used the same regression model as above. We performed Skat, Zeggini-Burden test, and a combined (skat and burden) SkatO test. Variants were grouped into genes based on UCSC ref-flat file for GRCH38 combining all possible exons to one canonical transcript per gene. We restricted them to variants with a minor allele frequency of 5% or below in this dataset and missense according to VEP (*n* = 79,218). Specifically, we included missense, frameshift, stop gained, stop lost including missense frameshift stop gained, or lost caused by splicing (Appendix A). To test rare variant associations between biologic pathways and KC, we combined selected individual genes to panels (Appendix A). Because of the increased number of variants per panel, we restricted this analysis to a combined Burden-Kernel test (SkatO).

Quantile-quantile plots and Manhattan plots were produced using qqman-R package, regional plots were produced with locus zoom, and LD correlation values were derived from European ancestries [12]. Regional plots to visualize rare variant association were produced with R-package Gviz [13].

## 3. Results

### 3.1. Patient Characteristics

Demographic data of the study participants (*n* = 150; cases/controls *n* = 75 each) and pigmentation phenotype are shown in Table 1. Sixty pairs (80%) were male, 15 pairs (20%) were female, with a mean age of 71.45 ± 8.78 (cases) and 70.96 ± 8.37 (controls) years at the first visit and 13.88 ± 7.27 (cases) and 14.44 ± 9.00 (controls) years after TX. The mean (±SD) age at TX was 57.57 ± 11.32 (cases) and 56.52 ± 12.00 (controls) years. The majority of SOTR received a kidney (*n* = 70, 47%), 37% (*n* = 56) received a heart, 11% received a lung (*n* = 16), and 5% a liver (*n* = 8). The average first occurrence of KC was 5.63 ± 4.80 years after TX (Table 1).

The majority of study participants had FST II (72, 48%) or FSTIII (74, 49%), while four individuals (3%) had FST I. Cases had more FST II (55% vs. 41% in controls) and controls had more FST III (42% vs. 32% in cases), but the distribution of FST between cases and controls was not significantly different (*p* = 0.25). A total of 1335 histologically verified KC were analyzed in this study: 44% were BCC (*n* = 593), 38% were cSCC/BowCa (*n* = 506), and 18% were in situ carcinomas (*n* = 236). On average, we observed 7 cSCC/BowCa, 8 BCC, and 3 BD per individual case (Table 1). Cumulative UV exposure and skin aging (modified SCINEXA score) were comparable in this subset of the cohort (Table 1).

### 3.2. Common Variants

We evaluated the association between 81,919 common variants (MAF > 5%) and the number of KC in 150 matched SOTR (Appendix A). We found one genome-wide significant association of an SNP located in the *EXOC3* (rs72698504) gene (Figure 1A,C and Table 2). An overview of the most significant common variants associated with the number of KC is given in Table 2. We did not observe significant inflation of the test statistics (Figure 1B). For the sensitivity analysis, we dichotomized the outcome, which did not return any genome-wide significant results. In addition, we regressed the genotype data against a skin aging score, which also did not return any significant results.

Upon closer investigation of the genes carrying common SNPs, we found clues as to how they may contribute to KC development and progression. EXOC3, a part of the exocyst complex, together with ZNF641 (Table 2), has recently been identified as a susceptibility locus for Barrett esophagus and esophageal adenocarcinoma [14]. The exocyst complex alongside WASHC1 is also involved in intracellular protein trafficking, facilitating invasive cell migration and resulting in matrix degradation [15]. Similarly, THBS2 mediates cell-to-cell and cell-to-matrix interactions and promotes tumor progression [16,17,18].

WASHC1 also participates in immune regulation via recycling of the receptors expressed on the surface of immune cells [19,20]—both mechanisms are thought to promote KC development. Further immunomodulation could be caused by UNC13A, which facilitates vesicle release and positively regulates TNF-α secretion from activated macrophages [21], thereby decreasing cytotoxic T-cell activity and increasing tumor growth [22].

In our study, we found genetic variation in two non-coding RNAs (PITRM1-AS1 and THBS2-AS1, Table 2), thus underlining the findings of a recent study by Khan and colleagues investigating the role of non-coding RNA dysregulation in skin cancer [23]. For example, PITRM1-AS1 has recently been shown to interact with KLF6, a transcription factor and tumor suppressor dysregulated in several cancers [24,25].

### 3.3. Rare Variants

A total of 255,127 high quality rare variants were present in our dataset (*n* = 150), of which 177,296 were synonymous variants (Appendix A). Variants assigned, missense, frameshift, and stop lost or gained by the Ensembl variant effect predictor were combined to a call set of 77,831 variants and then further restricted to 57,440 with an allele frequency below 5% in this dataset. Variants were then grouped according to NCBI RefSeq genes according to their genomic position, yielding a total of 14,744 genes with one or more missense variants in this dataset. We then performed a Burden and a combined kernel and burden test (SkatO) [26]. Rare variants significantly associated with KC across several association test methods are shown in Table 3 and Figure 2.

Several of the genes described here can be assigned to categories that are involved in the development of solid tumors: *MC1R* is involved in pigmentation/UV protection [27,28], *EPB41* [29,30,31,32,33,34] and *MYCT* [35,36,37,38] in tumor suppression, and *ADGRG3* in immunomodulation [39,40,41,42]. Studies on MGME1 suggest a direct involvement in cancer development and progression [43,44], and finally upregulation of the EPH8A gene expression was associated with a poor prognosis in ovarian, oral tongue, and gastric cancers [45,46,47].

By grouping several genes into panels (pigmentation, cellular response to UV, hereditary cancer, base excision repair, DNA double strand break repair, and immunosurveillance) and testing rare variant associations, we identified the gene panel “cellular response to UV” significantly associated with the development of multiple KCs.

## 4. Discussion

Long-term immunosuppressed SOTR are at increased risk for developing KC [4]. The driving force for the development of KC is UV radiation [4,48], including sun exposure prior to organ transplantation [4]. KC risk also increases with age at TX and duration, level, and type of IS therapy. Therefore, regular dermatologic care is a crucial component of SOTR follow-up, especially for SOTR, with an already increased risk of skin cancer [49].

Several KC risk assessment tools have been used to date [50]. Recently, SUNTRAC (Skin and UV Neoplasia Transplant Risk Assessment Calculator [51]), an easy-to-use online screening tool was validated in a large European organ transplant cohort, but without considering the genetic background of the patients [52]. The predisposition to develop skin cancer has been shown to aggregate in families [53] and to be found in known cancer-related genes [54,55]. Moreover, GWAS-derived associations between SNPs and patients with KC have been combined into polygenic risk scores for individual risk prediction of skin cancer [56,57,58,59,60,61]. In a recent GWAS by Sarin et al., it was estimated that 25% of the cSCC risk using all common variants in their dataset and 8% of the variance in their cSCC risk phenotype was explained by 22 genome-wide significant hits. This suggests the presence of additional, yet to be discovered, genetic loci that modulate cSCC risk.

Our study overcomes two major limitations of many GWAS studies: (1) our sample collection benefits from a deep phenotype that allows for very precise case-control matching. (2) We evaluated genetic variation with WES, allowing us to examine the contribution of rare variants to KC risk. In our cohort, we recorded a total of 1335 histologically verified KCs in 75 organ transplanted patients. We meticulously matched those cases with SOTRs in whom KC did not occur, taking into account not only the age at TX, type of organ, type and duration of IS therapy, and time after TX, but also focusing on their UV exposure history, thereby creating a unique cohort that allowed us to identify robust genetic associations to KC risk in a relatively small sample collective.

On the other hand, there are several limitations in our study, with the sample size representing the most significant constraint. However, we would like to emphasize that the sample population of organ transplant patients developing KC is quite small, making it challenging to find suitable patients for our study. Consequently, recruiting patients for this study required more than 3 years. Additionally, conducting whole exome analysis incurs certain costs, which is why our sample size is relatively small, limiting our analysis options due to reduced statistical power. Based on our power analysis (see Appendix A), we determined that we can achieve 90% power in our study with approximately 150 participants, even for variants with a minor allele frequency as low as 0.1. However, it is important to note that our analysis of common SNPs is limited to the coding region of the human genome due to the utilization of the exome analysis. This distinction makes our study less directly comparable to published studies that utilize genotyping array technology, which has a broader coverage across the entire genome. Another limitation of our study is the heterogeneity of cancer entities including post-transplantation variations However, within the cohort we examined, we encountered very few patients who had only one specific type of KC. Specifically, we identified two patients with only basal cell carcinoma (BCC) and eight patients with exclusively cutaneous squamous cell carcinomas (cSCCs) with or without actinic damage (BD). This limited number of cases would have made it nearly impossible to conduct a stratified analysis. Additionally, a recent review by Choquet et al. on the genetic risk factors of KC revealed several similarities in the genetic makeup of cSCC and BCC.

We evaluated the association between 81,919 common variants (MAF > 5%) and the number of KCs, to identify one genome-wide significant association of an SNP located in the *EXOC3* (rs72698504) gene (Figure 1A,C and Table 2). EXOC3 is one of the components of the exocyst complex that targets vesicles for exocytosis and is involved in various physiological processes, including cell migration [62,63], autophagy [64], cell cycle [65,66], and apoptosis [67,68]. Several components of the exocyst complex are associated with carcinogenesis in various cells and tissues. Recently, *EXOC3* was identified as a susceptibility locus for Barrett esophagus and esophageal adenocarcinoma [14].

Aside fro, *EXOC3,* the most significant SNP associated with multiple KCs, we found SNPs in the following genes *CYB561*, *WASHC1*, *PITRM1*-*AS1*, *MUC8*, *UNC13A*, *IL26*, *ABI3BP*, *ZNF641*, and *THBS2*-*AS1*. *WASHC1* is a member of the Wiskott−Aldrich syndrome protein (WASP) family, which is involved in the regulation of actin polymerization, endosomal protein sorting, and trafficking, as well as in invasive cell migration [15,19,69,70,71,72,73,74]. It cooperates with the exocyst complex in matrix degradation via exocytosis of trans-membrane type 1 matrix metalloproteinase (MT1-MMP) at invadopodia [15].

Long non-coding RNAs (lncRNAs) such as pitrilysin metallopeptidase 1 (PITRM1)-AS1 are RNA molecules that are longer than 200 nucleotides and do not encode proteins. They often act as tumor inhibitors in case of PITRM1-AS1 by regulating autophagy and epithelial–mesenchymal transition [25]. PITRM1-AS1 has recently been shown to interact with KLF6, a transcription factor and tumor suppressor that is dysregulated in several cancers [24]. ABI gene family member 3-binding protein (ABI3BP) is involved in the regulation of actin cytoskeleton reorganization and cell movement and has been reported to act as a tumor suppressor [75,76].

We think that the newly discovered SNPs could be included in polygenic risk scores to stratify SOTR with respect to genetic risk for developing KC. Especially because most of the SNPs discovered in our study have not been previously associated with the development of KC. On the other hand, we also observed overlaps between SNPs discovered in our research and previously published data. These overlapping SNPs were found in genes related to skin pigmentation, such as melanocortin-1 receptor (MC1R), tyrosinase (TYR), and solute carrier family 45 member 2 (SLC45A2). This suggests that these genes may contribute to increased sensitivity to ultraviolet (UV) radiation and an elevated risk of keratoconus (KC) [1,2]. Furthermore, genome-wide association studies (GWAS) have identified specific genetic loci associated with KC susceptibility. These loci are connected to genes involved in keratinocyte differentiation, cancer development and progression, DNA damage repair, immune regulation, and pigmentation [1,3,4,5,6,7,8,77].

Our study setting also allowed us to assess the genetic contribution of rare variants to KC risk in organ transplant recipients. We found more rare variants than expected within the *MC1R* exonic regions. MC1R is one of the key regulators of skin pigmentation and thus is important for protecting the skin from UV damage. Mutations in the MC1R gene are known to lead to decreased eumelanin production, which impairs UV protection and thus increases susceptibility to skin cancer [27,28]. This association with multiple KCs confirms our previous finding that carriers of specific *MC1R* variants had a higher number of KCs even in individuals with darker skin types, suggesting a pigmentation-independent effect of *MC1R* variants [5].

In addition to *MC1R*, we identified several genes that had a high mutational burden compared with the tumor-free controls. Based on our analysis, we think that *EPHA8*, *EPB41, EPO*, *MYCT1*, *ADGRG3,* and *MGME1* (Appendix A and Figure 2) may be involved in the pathogenesis of KC.

Erythrocyte membrane protein band 4.1 (EPB41) links cell membrane proteins to the cytoskeleton and is involved in many cellular processes such as cell proliferation, cytokine secretion, cell adhesion, and cell migration [29,30,31,32,33,34]. EPB41 suppresses epidermal growth factor receptor (EGFR) activation [29] and thus plays an important role as a tumor suppressor in cancer development. Dysregulation and excessive activation of EGFR leads to the hyperproliferation of many cell types such as epidermal keratinocytes and thus has been linked to tumorigenesis in SCC [78,79]. Moreover, EPB41 expression serves as a prognostic biomarker for various tumors, as EGFR overactivity is associated with a poor prognosis [80,81,82].

MYCT1, a direct c-Myc target gene, regulates numerous downstream genes and inhibits epithelial−mesenchymal transition, migration, and apoptosis in various cancer cells, suggesting that it acts as a tumor suppressor [35,36,38,83].

EPHA8 belongs to the erythropoietin-producing hepatocellular receptor family of transmembrane receptor tyrosine kinases, which are dysregulated in various cancers. In colon cancer and gliomas, *EPHA8* has been reported to be targeted by miR-10a, resulting in the inhibition of cell invasion and migration [45,46,47]. 

Finally, to gain insight into the pathways involved in KC development, we grouped several genes into panels (Appendix A) to test them for their association with multiple KCs. We found that the ”cellular response to UV” gene panel was significantly associated with KC development. This finding underscores that UV exposure is one of the driving forces in the pathogenesis of KC and that genetically determined interindividual variability in the cellular response to UV may be reflected in predisposition to KC.

## 5. Summary

In this study, despite the relatively small sample size, we identified a set of eight novel SNPs (*p* < 10^−5^) that increase the risk of keratinocyte cancer and a set of eight genes carrying rare variants that are robustly associated with KC development. For both the rare and common variants, closer examination of the genes carrying these novel variants revealed that they are primarily involved in pigmentation/UV protection, immunomodulation, and tumor suppression. We suggest using KC risk assessment tools such as SUNTRAC to include the genetic background of patients. However, larger cohorts of organ transplant patients are needed to create a robust genetic risk score tailored to KC risk stratification of organ transplant recipients.

## Figures and Tables

**Figure 1 cancers-15-03354-f001:**
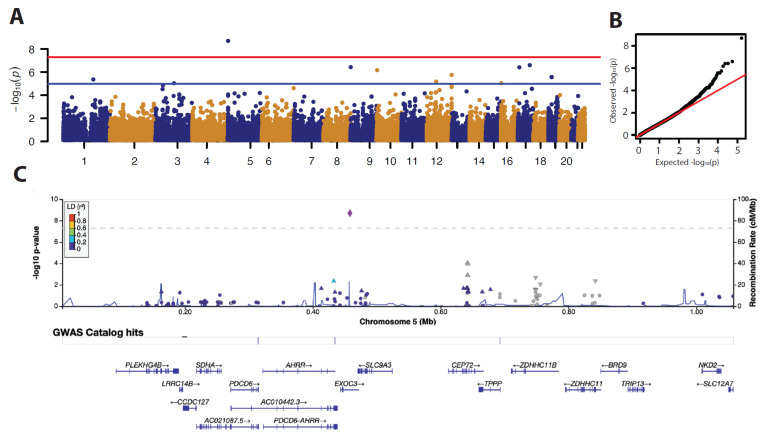
Overview of association results between the number of KCs per patient and their common variants. (**A**) Manhattan plot. The x axis shows chromosomes, the y axis shows *p*-values for the SNP association to KC. (**B**) Qq plot comparing *p*-values expected by chance to the actual observed *p*-values observed in this study. (**C**) Locus zoom plot providing information of SNPs adjacent to the lead SNP in purple located in the *EXOC3* gene. Dots are colored according to their linkage disequilibrium relative to the lead SNPs measured in European ancestries.

**Figure 2 cancers-15-03354-f002:**
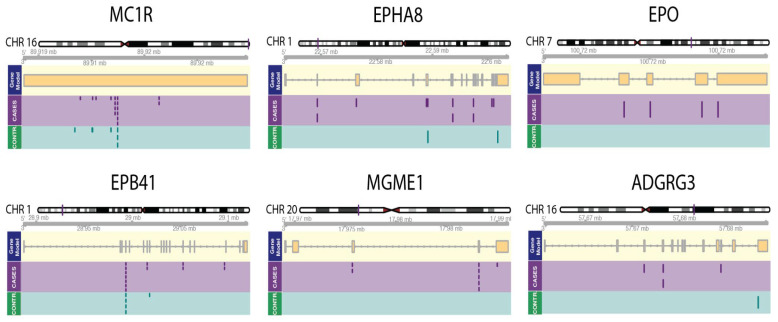
Visualization of groupwise tests. In each panel (gene), we plotted, from top to bottom, an ideogram, then chromosomal position on the chromosome. Next is the gene model showing all exons and introns of the analyzed gene. The panel labeled cases gives all rare variants of patients that developed KC in this cohort (*n* = 75). The green panel labeled CONTR. gives all rare variants present in patients that did not develop KC.

**Table 1 cancers-15-03354-t001:** Patient characteristics.

	Total	Cases	Controls	*p*-Value
	n = 150	n = 75	n = 75	
Gender, n (%)				
Male	120 (80)	60	60	*p* = 1.00
Female	30 (20)	15	15	*p* = 1.00
Type of organ transplant recipient, n (%)				
HTR	56 (37)	28	28	*p* = 1.00
KTR	70 (47)	35	35	*p* = 1.00
LTR	8 (5)	4	4	*p* = 1.00
LuTR	16 (11)	8	8	*p* = 1.00
Age at visit, years (mean ± SD)	71.21 ± 8.56	71.45 ± 8.78	70.96 ± 8.37	*p* = 0.73
Age at TX, years (mean ± SD)	57.05 ± 11.64	57.57 ± 11.32	56.52 ± 12.00	*p* = 0.68
Post-TX period, years (mean ± SD)	14.16 ± 8.16	13.88 ± 7.27	14.44 ± 9.00	*p* = 0.68
Time to 1st post-TX KC, years (mean ± SD)		5.63 ± 4.80		
IS-regimen, n (%)				
Steroids Yes	83 (55)	38 (51)	45 (60)	*p* = 0.25
No	67 (45)	37 (49)	30 (40)	
mTOR inhibitors Yes	30 (20)	18 (24)	12 (16)	*p* = 0.22
No	120 (80)	57 (76)	63 (84)	
Antimetabolites Yes	114 (76)	52 (69)	62 (83)	*p* = 0.06
No	36 (24)	23 (31)	13 (17)	
Calcineurin inhibitors Yes	123 (82)	61 (81)	62 (83)	*p* = 0.83
No	27 (18)	14 (19)	13 (17)	
Azathioprine Yes	5(4)	2 (3)	3 (4)	*p* = 0.65
No	145 (97)	73 (97)	72 (96)	
Belatacept Yes	2 (1)	1 (1)	1 (1)	*p* = 1.00
No	148 (99)	74 (99)	74 (99)	
Total number of KC (KC: SCC + BowCa/BCC/BD) Number of KC per case (mean ± SD) cSCC + BowCa BCC BD		1335: 506/593/236 18 ± 22 7 ± 12 8 ± 11 3 ± 4		
Fitzpatrick Skin Type, n (%) I II III	4 (3) 72 (48) 74 (49)	2 (3) 41 (55) 32 (43)	2 (3) 31 (41) 42 (56)	*p* = 0.25
Cumulative UV exposure, hours (mean ± SD)	47,111 ± 27,349	45,811 ± 24,595	48,410 ± 29,962	*p* = 0.56
Modified SKINEXA skin aging score Score (mean ± SD)	20 ± 7	20 ± 7	20 ± 7	*p* = 1.00

HTR = heart transplant recipient; KTR = kidney transplant recipient; LTR = liver transplant recipient; LuTR = lung transplant recipient; TX = transplantation; KC = keratinocyte cancer; cSCC = cutaneous squamous carcinoma; BowCa = Bowen’s carcinoma; BCC = basal cell carcinoma; BD = Bowen’s disease; SD = standard deviation; IS = immunosuppression.

**Table 2 cancers-15-03354-t002:** Common variants associated with KC.

rsID	Symbol	Chromosome	Position	Ref	Alt	Effect	SE	*p*-Value	Impact	Biotype
rs72698504	*EXOC3*	5	458,036	C	T	25.13	4.19	1.98 × 10^−9^	modifier	protein_coding
rs4968774	*CYB561*	17	63,438,051	G	T	18.08	3.50	2.50 × 10^−7^	modifier	protein_coding
rs200377821	*WASHC1*	9	14,816	C	G	24.21	4.76	3.71 × 10^−7^	moderate	protein_coding
rs4239111	*SCARNA21*	17	7,908,680	T	C	−14.33	2.82	3.86 × 10^−7^	modifier	scaRNA
rs76088740	*PITRM1-AS1*	10	3,151,220	C	T	23.70	4.77	6.76 × 10^−7^	modifier	lncRNA
rs76373320	*MUC8*	12	132,474,266	G	C	23.15	4.85	1.79 × 10^−6^	modifier	CTCF_binding_site
rs76225638	*UNC13A*	19	17,627,849	C	T	19.25	4.10	2.68 × 10^−6^	modifier	protein_coding
rs10748100	*IL26*	12	68,201,939	T	C	13.71	3.32	3.57 × 10^−5^	low	protein_coding
rs34999788	*ABI3BP*	3	100,898,824	G	T	13.75	3.10	9.27 × 10^−6^	low	protein_coding
rs751315	*ZNF641*	12	48,340,492	C	T	14.64	3.64	5.71 × 10^−5^	modifier	protein_coding
rs35631991	*THBS2-AS1*	6	169,237,752	G	A	20.40	4.84	2.49 × 10^−5^	modifier	lncRNA

Symbol is the gene name of the nearest gene to the associated SNP, followed by chromosome, chromosomal position, reference, and altered allele. Effect reflects the coefficient of the regression analysis SE the standard error, impact, and biotype were provided by the Ensembl variant effect predictor.

**Table 3 cancers-15-03354-t003:** Rare variants associated with KC.

Gene	NumPolyVar	Beta_CMC-Wald	SE CMC-Wald	*p*-Value CMC-Wald	*p*-Value SkatO	*p*-Value Skat	*p*-Value Burden_Zeggini
*MC1R*	11	16.25	3.78	1.67 × 10^−5^	5.04 × 10^−8^	2.53 × 10^−7^	7.73 × 10^−7^
*EPHA8*	10	13.82	5.69	1.51 × 10^−2^	2.16 × 10^−10^	4.49 × 10^−12^	3.56 × 10^−7^
*EPB41*	5	19.89	4.36	5.17 × 10^−6^	1.34 × 10^−6^	2.64 × 10^−6^	1.48 × 10^−6^
*EPO*	4	54.75	7.77	1.85 × 10^−12^	2.15 × 10^−10^	4.09 × 10^−15^	1.01 × 10^−9^
*MYCT1*	4	29.48	4.36	1.42 × 10^−11^	1.19 × 10^−8^	1.03 × 10^−7^	3.31 × 10^−9^
*ADGRG3*	4	32.51	8.51	1.34 × 10^−4^	2.16 × 10^−10^	4.98 × 10^−13^	4.13 × 10^−8^
*MGME1*	4	30.35	5.95	3.40 × 10^−7^	3.74 × 10^−7^	5.15 × 10^−7^	2.50 × 10^−6^
*ZNF276*	9	17.62	4.05	1.35 × 10^−5^	4.07 × 10^−8^	1.42 × 10^−8^	4.11 × 10^−5^

NumPolyVar reflects the number of polymorph variants in the analyzed gene, Beta CMC-Wald is the coefficient calculated from a Wald ratio test comparing rare variants in cases to rare variants in controls. SE is the standard error. *p*-values are calculated with several methods: a Kernel method Skat(*p*-ValueSkat), two Burden test CMC-Wald and Zeggini (Beta_CMC-Wald, SE CMC-Wald, *p*-Value CMC-Wald, *p*-Value Burden_Zeggini) and a method combining Burden and Kenel tests called Skat-O (*p*-Value SkatO).

## Data Availability

Summary statistics for SNPs > 5% are available via figshare: https://doi.org/10.6084/m9.figshare.22494538.v1 (accessed on 20 June 2023) FASTQ, VCF or BAM (GRCh38) files will be made available upon reasonable request.

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
