# Peer review of "Identification of Genetic Risk Factors for Keratinocyte Cancer in Immunosuppressed Solid Organ Transplant Recipients: A Case-Control Study"

_cancers, 2023, doi:10.3390/cancers15133354_

Round 1

Reviewer 1 Report

Thank you for the opportunity to review this manuscript, in which Sunder-Plassmann et al undertake a case-control study to explore genetic risk factors for keratinocyte cancer in organ transplant recipients. Keratinocyte cancer is the most common malignancy seen post-transplant, and therefore there is an unmet need to identify novel methods to risk stratify patients to guide surveillance and intervention. This is, therefore, an important issue in the field.

The manuscript is well-written, appropriately referenced, and easy to follow, and the authors should be congratulated on this. I do, however, have a few concerns and suggestions which should be addressed prior to publication.

MAJOR COMMENTS:

- The authors treat all keratinocyte cancer as a single pathology in this study - however basal cell carcinoma and squamous cell carcinoma have markedly different pathogeneses and behaviours post-transplant - more subtle genetic influences may be lost in this combined approach. The authors do not appear to acknowledge this limitation in their work. Is there any scope to reanalyse the data subdivided by type of KC? Indeed, there appears to be no acknowledgement of any study limitations in the discussion and this should be included.

- It is interesting that patients (at an individual level) appear to have had more BCC than CSCC when CSCC is usually more prevalent in transplant recipients - why is this?

- The authors do not discuss any of the pre-existing studies in this field, which have identified various SNPs associated with increased risk of subsets of KC, or cross-reference with their findings. How do their findings compare? This should be included in the discussion.

MINOR COMMENTS:

- The authors move between using 'KC' and 'NMSC' at various points in the manuscript (e.g. KC on page 1, and NMSC on page 3)- this should be consistent throughout.

- Details of the cohort size is duplicated in first line of the methods and again in the first line of the results. Insertion into the results is probably more appropriate, but in the methods the authors should discuss how this cohort size was decided up - was there a power calculation?

- It would be helpful to include the study design in the title to guide the reader - "Identification of genetic risk factors for Keratinocyte Cancer in immunosuppressed solid organ transplant recipients: a case-control study" would be more informative.

Author Response

Please see pdf attached

Reviewer 2 Report

The authors should be congratulated for this manuscript, based on a well-conducted and complicated study. A few comments and questions:

1. Please amend the last paragraph of the Introduction, regarding the patients, as this information is and should be provided in the Methods section.

2. Why the number of KC was used as a continuous variable for the regression models, when the groups are KC+/-? Is there evidence that a SOTR patient has a different genotype with 8, 10 or 15 KCs?

3. No limitations of the study are provided, apart from the sample size, which is mentioned in the Summary instead of the Discussion.

4. What is the clinical value of this study? Although the authors state that "the newly discovered SNPs could be included in polygenic risk scores to stratify SOTR with respect to genetic risk for developing KC" no relevant studies are discussed.

5. No sample size calculation is reported.

6. Why the cut-off point of 5 KCs was chosen for the cases to be included in the study? It is well known that SOTRs have >100 times the risk of developing SCC than non-transplant patients, so please comment on this study's population characteristics.

7. The cases-controls seem extremely well-matched. How was the recruitment (of the controls) done?

8. Based on this study results, please provide recommendations for further research.

Author Response

Please see pdf attached.

Round 2

Reviewer 1 Report

My thanks to the authors for their incorporation of changes. They should be congratulated on their work.